# HutCRS: Hierarchical User-Interest Tracking for Conversational Recommender System

**Mingjie Qian**[1*]**, Yongsen Zheng**[1*]**, Jinghui Qin**[2] and **Liang Lin**[1†]

[1]Sun Yat-sen University, [2]Guangdong University of Technology

{z.yongsensmile, scape1989}@gmail.com

qianmj7@mail2.sysu.edu.cn, linliang@ieee.org

## Abstract

Conversational Recommender System (CRS) aims to explicitly acquire user preferences towards items and attributes through natural language conversations. However, existing CRS methods ask users to provide explicit answers (yes/no) for each attribute they require, regardless of users' knowledge or interest, which may significantly reduce the user experience and semantic consistency. Furthermore, these methods assume that users like all attributes of the target item and dislike those unrelated to it, which can introduce bias in attribute-level feedback and impede the system's ability to accurately identify the target item. To address these issues, we propose a more realistic, user-friendly, and explainable CRS framework called Hierarchical User-Interest Tracking for Conversational Recommender System (HutCRS). HutCRS portrays the conversation as a hierarchical interest tree that consists of two stages. In stage I, the system identifies the aspects that the user prefers while the system asks about attributes related to these positive aspects or recommends items in stage II. In addition, we develop a Hierarchical-Interest Policy Learning (HIPL) module to integrate the decision-making process of which aspects to ask and when to ask about attributes or recommend items. Moreover, we classify the attribute-level feedback results to further enhance the system's ability to capture special information, such as attribute instances that are accepted by users but not presented in their historical interactive data. Extensive experiments on four benchmark datasets demonstrate the superiority of our method. The implementation of HutCRS is publicly available at https://github.com/xinle1129/HutCRS.

## 1 Introduction

Conversational recommender systems (CRSs) have become one of the trending research topics in re-

---

[*]Both authors contributed equally to this research.
[†]Corresponding author.

cent years(Lei et al., 2020a), utilizing the user's online feedback to conduct dynamic and explainable recommendations through interactive conversations with the user. One of the challenges for CRS is how to efficiently acquire user preferences and quickly narrow down the recommendation candidates. In this regard, attribute-based CRS has been extensively studied, as whether a user likes an attribute can significantly reduce the recommendation candidates(Gao et al., 2021).

Among the different proposed problem settings for attribute-based CRS, the single-round(Christakopoulou et al., 2018; Sun and Zhang, 2018) setting is the earliest to be introduced. However, this setting is impractical in real-world deployments because the CRS only makes recommendations once, ending the session regardless of the results. To address this issue, the multi-round(Lei et al., 2020a,b; Xu et al., 2021; Ren et al., 2021; Deng et al., 2021; Tu et al., 2022; Hu et al., 2022; Zhang et al., 2022) setting has been proposed, wherein the CRS interacts with the user by asking attributes and recommending items multiple times until the task succeeds or the user terminates the session. Multi-round Conversational Recommendation(MCR) aims to make successful recommendations with fewer conversation turns.

Although MCR is widely acknowledged as the most realistic setting for CRS, the existing MCR assumptions regarding user interest deviate from real-world scenarios. First, MCR presupposes that users have definitive answers(yes or no) for all queried attributes. However, users might be unaware of or indifferent to certain attributes, and compelling them to respond could reduce the user experience and generate inconsistent conversation utterances. Additionally, MCR assumes that users will accept all attributes that belong to the target item and reject those unrelated to it. In practice, users might not necessarily like all attributes of the target item, nor dislike attributes that are not

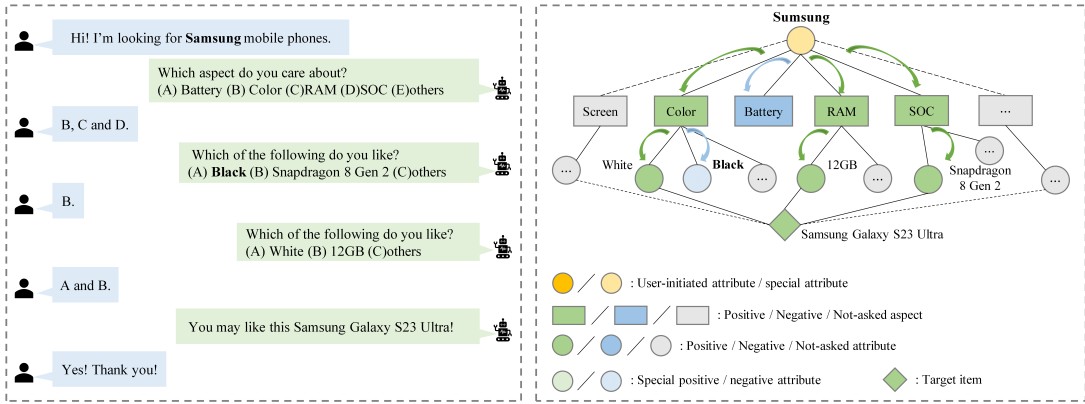

Figure 1: An illustration of conversation in HutCRS. The hierarchical interest tree (right side) based on the user's feedback enables the system to better understand the needs and preferences of the user.

included in the target item. This discrepancy can result in invalid attribute-level feedback and prevent the system from identifying the target item accurately. For instance, considering a target item "an esports phone with a large battery capacity but low-performance camera", the user doesn't prioritize the performance of the camera, but it doesn't imply that the user prefers a low-performance camera. If the system asks about the "low-performance camera", it may receive negative feedback which could cause it to miss out on the target item.

To address the aforementioned issues, we propose a novel framework **H**ierarchical **U**ser-Interest **T**racking for **C**onversational **R**ecommender **S**ystem (HutCRS). As shown in Figure 1, we represent the user's preferences in the current session as a hierarchical tree that depicts the user's interests and priorities, facilitating a better understanding of their needs and preferences. Specifically, the system first prompts the user to specify their aspects of interest based on user-initialized attributes. Once this information is obtained by the system, it will either ask about attributes associated with the positive aspects or recommend items. The interaction based on the hierarchical interest tree enables the system to ask about attributes that the user is indeed interested in, thereby improving user experience and consistency in the conversation. Besides, when attribute instances belong to the target item and align with the aspects of interest to the user, they are more likely to be accepted. This clear preference will lead to the generation of more authentic feedback. Inspired by Zhang et al. (2022), we develop a Hierarchical-Interest Policy Learning (HIPL) module to integrate the decision-making process

of which aspects to ask and when to ask attributes or when to recommend items. In addition, we classify the attribute-level feedback results to further enhance the system's ability to capture special information. Positive attributes that are not present in the historical data and negative attributes that are present in the historical data are marked as "special". Extensive experiments on Yelp, LastFM, Amazon-Book, and MovieLen demonstrate the superiority of HutCRS compared with state-of-the-art baselines.

In summary, our contributions can be concluded as follows:

- We extend existing CRS to a more realistic, user-friendly, and explainable setting.
- We propose the hierarchical interest tree and utilize the classification results of attribute-level feedback to further enhance the system's ability to comprehend the needs and preferences of users.
- Experiments on four benchmarks show the superiority of HutCRS compared with state-of-the-art baselines.

## 2 Related Works

CRS is a recommendation system that can elicit the dynamic preferences of users and take actions based on their current needs through real-time multiturn interactions(Gao et al., 2021). In this work, we focus on attribute-based CRS(Sun and Zhang, 2018; Lei et al., 2020a,b; Deng et al., 2021; Zhang et al., 2022) which ask users attributes to efficiently acquire user preferences and quickly narrow down the recommendation candidates. The problem settings of attribute-based CRS can be mainly divided into two categories: single-round and multi-round.

In single-round scenario(Christakopoulou et al., 2018; Sun and Zhang, 2018), the CRS only makes recommendations once, ending the session regardless of the results. In multi-round scenario, the CRS can interacts with the user by asking attributes and recommending items multiple times until the task succeeds or the user terminates the session.

Due to the wide acknowledgement of Multi-round Conversational Recommendation (MCR) as the most realistic setting(Lei et al., 2020a) for CRS, this work is conducted based on the multi-round scenario. EAR(Lei et al., 2020a) extends the single-round setting into a multi-round setting where the system can recommend multiple times. SCPR(Lei et al., 2020b) models the MCR as an interactive path reasoning problem on a graph, which is able to prune off many irrelevant candidate attributes. FPAN(Xu et al., 2021) utilizes a gating mechanism to aggregate online feedback from users. UNICORN(Deng et al., 2021) proposes to formulate three decision-making problems in CRS as a unified policy learning task based on dynamic weighted graph. MIMCPL(Zhang et al., 2022) extends MCR to a more realistic scenario setting named MIMCR where the user may accept multiple attribute instances with the same type and multiple items with partially overlapped attributes. However, the existing MCR assumptions about user interest still deviate from real-world scenarios. Hence, we propose a novel framework called HutCRS to tackle these issues.

## 3 Definition and Preliminary

To better describe and understand the HutCRS, we set up some basic notions in this section. We define the sets of users and items as $\mathcal{U}$ and $\mathcal{V}$, respectively. Additionally, we separately define the sets of aspect (i.e. type) and attribute instances as $\mathcal{C}$ and $\mathcal{P}$. Each item instance $v \in \mathcal{V}$ is associated with a set of attribute instances $\mathcal{P}_v$. Each attribute instance $p \in \mathcal{P}$ has its corresponding aspect $c_p \in \mathcal{C}$. In each episode, there is a set $\mathcal{V}_u$ of items that are acceptable to the user $u \in \mathcal{U}$. The set is represented as $\mathcal{V}_u = \{v_1, v_2, \cdots v_{N_v}\}$, where $N_v$ is the number of acceptable items, $\mathcal{P}_{v_1} \cap \mathcal{P}_{v_2} \cap \cdots \cap \mathcal{P}_{v_{N_v}} = \mathcal{P}_{same} \neq \emptyset$ and $\mathcal{P}_i \neq \mathcal{P}_j$. As shown in Figure 1, a session starts with a preferred attribute instance $p_0 \in \mathcal{P}_{same}$ specified by user $u$. Then, the agent asks about which aspects the user prefers from the candidate aspect set $\mathcal{C}_{cand}$ until obtaining positive feedback. In the remaining turns, it will repeatedly ask about attributes associated with the positive aspects or recommend items until at least one acceptable item is successfully recommended to the user or the system reaches the maximum number of turn $T$.

## 4 HutCRS

To improve user experience and semantic consistency, we propose a novel framework Hierarchical User-Interest Tracking for Conversation Recommender System (HutCRS) for MCR. Similarly to Zhang et al. (2022); Lei et al. (2020b); Deng et al. (2021), our framework also aims to learn the policy network $\pi(a_t|s_t)$ to maximize the expected cumulative rewards as: $\pi^* = \arg\max_{\pi \in \Pi} \mathbb{E}[\sum_{t=0}^{T} r_t]$, where $s_t$ denotes the current state, $a_t$ denotes the action taken by the agent and the $r_t$ is the intermediate reward. As shown in Figure 2, our framework portrays the conversation as a hierarchical interest tree that consists of two stages where each stage comprises two components: Hierarchical User Interest Tracking and Hierarchical-Interest Policy Learning. In stage I, the system aims to identify the aspects that the user prefers, while in stage II, the system will either ask about attributes related to these positive aspects or recommend items.

### 4.1 Hierarchical User Interest Tracking

In order to learn the policy network $\pi(a_t|s_t)$, it is necessary to characterize states and actions. To this end, we use the Hierarchical User State Tracking module to track the current state and the Interest-based Candidate Selection module to rank the candidates.

#### 4.1.1 Hierarchical User State Tracking

We utilize a hierarchical interest tree to represent the user's preferences in the current session. Unlike the approach presented in Zhu et al. (2018), our hierarchical interest tree does not necessitate extra time and resources for its construction. As shown in Figure 2, the full tree consists of the user-initialized attribute $p_0$ as the root node, all aspect instances as the first layer nodes, all attribute instances as the second layer nodes, and all items as the last layer nodes. As the interaction proceeds, we prune the full tree to obtain a dynamic tree based on user feedback. This dynamic tree only retains nodes that have been interacted with by the user in the current session, as well as those nodes that belong to the candidates. In addition, to enhance the system's ability to capture special infor-

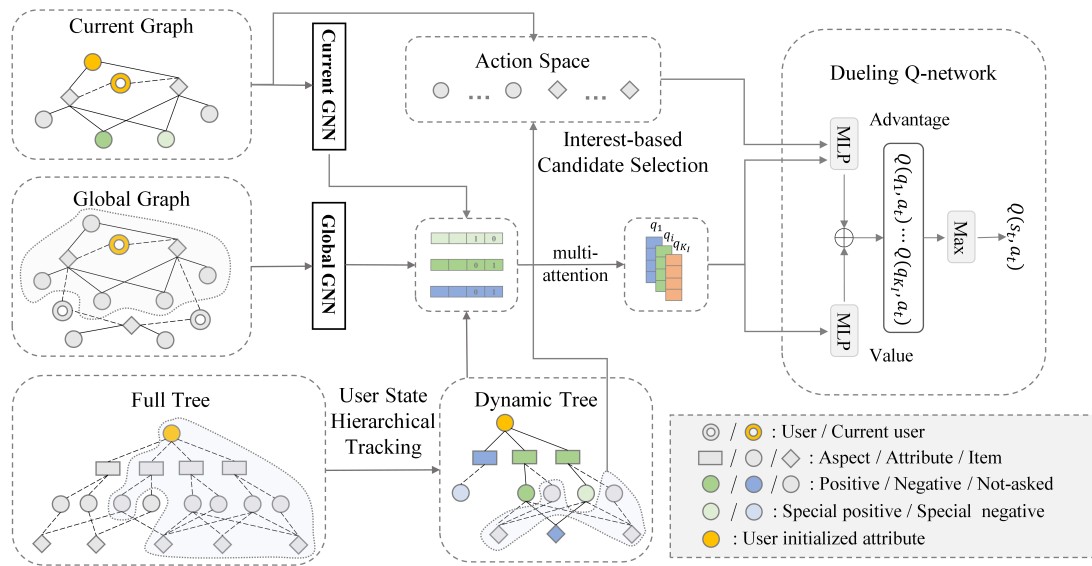

Figure 2: Overview of the proposed framework, HutCRS. HutCRS consists of two stages, each comprising two components: Hierarchical User Interest Tracking and Hierarchical-Interest Policy Learning.

mation, we also classify the attribute-level feedback results, where positive attributes $\mathcal{P}_{acc}$ not present in the historical data and negative attributes $\mathcal{P}_{rej}$ present in the historical data will be marked as "special". Hence, the current state includes four types of components: $s_t = \{u, \mathcal{C}_*^{(t)}, \mathcal{P}_*^{(t)}, \mathcal{V}_*^{(t)}\}$, where $\mathcal{C}_*^{(t)} = \{\mathcal{C}_{acc}^{(t)}, \mathcal{C}_{rej}^{(t)}, \mathcal{C}_{cand}^{(t)}\}$ denotes the state of aspects, and $\mathcal{P}_*^{(t)} = \Big\{\mathcal{P}_{acc}^{(t)}, \mathcal{P}_{rej}^{(t)}, \mathcal{P}_{special\_acc}^{(t)},$ $\mathcal{P}_{special\_rej}^{(t)}, \mathcal{P}_{cand}^{(t)}\Big\}$ denotes the state of attributes, $\mathcal{V}_*^{(t)} = \{\mathcal{V}_{rej}^{(t)}, \mathcal{V}_{cand}^{(t)}\}$ denotes the state of items.

*In stage I*, aspects that a user accepts or rejects in turn $t$ can be defined as $\mathcal{C}_{cur\_acc}^{(t)}$ and $\mathcal{C}_{cur\_rej}^{(t)}$ respectively. Before the system gets the accepted aspects $\mathcal{C}_{acc}$, some components are updated by $\mathcal{C}_{cand}^{(t+1)} = \mathcal{C}_{cand}^{(t)} - \mathcal{C}_{cur\_rej}^{(t)}, \mathcal{P}_{special\_rej}^{(t+1)} = \mathcal{P}_{special\_rej}^{(t)} \cup \mathcal{P}_{cur\_special\_rej}^{(t)}$. It's worth noting that we don't add $\mathcal{P}_{c_p \in \mathcal{C}_{cur\_rej}^{(t)}}$ to $\mathcal{P}_{cur\_rej}^{(t)}$ directly, but we get $\mathcal{P}_{cur\_special\_rej}^{(t)}$ from $\mathcal{P}_{c_p \in \mathcal{C}_{cur\_rej}^{(t)}}$. In this stage, we employ the *aspect instance-based union set strategy* to update $\mathcal{V}_{cand}^{(t+1)}$: $\mathcal{V}_{cand}^{(t+1)} = \{v | v \in \mathcal{V}_{p_0} \text{ and } \mathcal{C}_v \cap \mathcal{C}_{acc}^{(t)} \neq \emptyset\}$, where $\mathcal{V}_{p_0}$ is the item set in which all items are associated to $p_0$, and $\mathcal{C}_v$ is the aspect set in which all aspects are associated to $v$. Once the system gets the accepted aspects $\mathcal{C}_{acc}$, the system will transition to stage II.

*In stage II*, the candidate attribute set $\mathcal{P}_{cand}$ is initialized as $\mathcal{P}_{c_p \in \mathcal{C}_{acc}}$. Some components are updated by $\mathcal{P}_{cand}^{(t+1)} = \mathcal{P}_{cand}^{(t)} - \mathcal{P}_{cur\_acc}^{(t)} -$

$\mathcal{P}_{cur\_rej}^{(t)}, \mathcal{P}_{acc}^{(t+1)} = \mathcal{P}_{acc}^{(t)} \cup \mathcal{P}_{cur\_acc}^{(t)}, \mathcal{P}_{rej}^{(t+1)} = \mathcal{P}_{rej}^{(t)} \cup \mathcal{P}_{cur\_rej}^{(t)}, \mathcal{P}_{special\_acc}^{(t+1)} = \mathcal{P}_{special\_acc}^{(t)} \cup \mathcal{P}_{cur\_special\_acc}^{(t)}, \mathcal{P}_{special\_rej}^{(t+1)} = \mathcal{P}_{special\_rej}^{(t)} \cup \mathcal{P}_{cur\_special\_rej}^{(t)}$. In this stage, we employ the *attribute instance-based union set strategy* to update $\mathcal{V}_{cand}^{(t+1)}$. Unlike Zhang et al. (2022), we don't remove all items that contain the negative attributes, as in Lei et al. (2020b); Deng et al. (2021); Lei et al. (2020a): $\mathcal{V}_{cand}^{(t+1)} = \{v | v \in \mathcal{V}_{p_0} - \mathcal{V}_{rej}^{(t)} \text{ and } \mathcal{P}_v \cap \mathcal{P}_{acc}^{(t)} \neq \emptyset\}$.

### 4.1.2 Interest-based Candidate Selection

To reduce the action space in subsequent steps, we need to sort the candidate nodes in the dynamic tree. Following Deng et al. (2021), we construct a dynamic weighted graph to rank candidate items. Given a user $u$, we denote the dynamic graph at $t$-th turn as $\mathcal{G}_u^{(t)} = (\mathcal{N}^{(t)}, \mathcal{E}^{(t)})$:

$$\mathcal{N}^{(t)} = \{u\} \cup \mathcal{P}_{acc}^{(t)} \cup \mathcal{P}_{cand}^{(t)} \cup \mathcal{V}_{cand}^{(t)} \quad (1)$$

$$\mathcal{E}_{i,j}^{(t)} = \begin{cases} w_v^{(t)}, & \text{if } n_i = u, n_j \in \mathcal{V} \\ 1, & \text{if } n_i \in \mathcal{V}, n_j \in \mathcal{P} \\ 0, & \text{otherwise} \end{cases} \quad (2)$$

where $\mathcal{E}_{i,j}$ denotes the weighted edges between modes $n_i$ and $n_j$. $w_v^{(t)}$ is a scalar indicating the recommendation score of the item $v$ in the current state. To incorporate the special negative information, such $w_v^{(t)}$ is calculated as: $w_v^{(t)} =$

$\sigma\Big(e_u^\top e_v + \sum_{p\in\mathcal{P}_{acc}^{(t)}} e_v^\top e_p - \sum_{p\in\mathcal{P}_{rej}^{(t)}\cap\mathcal{P}_v^{(t)}} e_v^\top e_p - \sum_{p\in\mathcal{P}_{special\_rej}^{(t)}\cap\mathcal{P}_v^{(t)}} e_v^\top e_p\Big)$, where $\sigma(\cdot)$ is the sigmoid function, $e_u$, $e_v$ and $e_p$ are the embeddings of the user, item, and attribute, respectively.

Different from item ranking, those attributes with high scores should be able to not only capture the user's preferences but also better reduce the uncertainty of candidate items. Since HutCRS only asks about attributes related to the positive aspects which have already captured user's preferences, we utilize the entropy method(Wu et al., 2015) to rank the candidates.

## 4.2 Hierarchical-Interest Policy Learning

The purpose of this module is to decide *which aspects to ask and when to ask attributes or when to recommend items* to achieve successful recommendations in the fewest turns. We construct a current graph and a global graph to balance the user's short-term and long-term preferences. Moreover, we utilize the Hierarchical-Interest Extractor and Dueling Q-network to better understand users' hierarchical interests and decide the next action.

**Graph Representation.** For the current graph $\mathcal{G}_u^{(t)}$, which is introduced in 4.1.2, we employ a $L_c$-layer GCN(Kipf and Welling, 2016) to refine the node embeddings as in Zhang et al. (2022):

$$\mathbf{e}_n^{(l+1)} = \text{ReLU}\Big(\sum_{j\in\mathcal{N}_n^{(t)}} \frac{\mathbf{W}_c^{(l+1)}\mathbf{e}_j^{(l)}}{\sqrt{\sum_i \mathcal{E}_{n,i}^{(t)}\sum_i \mathcal{E}_{j,i}^{(t)}}} + \mathbf{e}_n^{(l)}\Big) \tag{3}$$

where $\mathbf{e}_n^{(l)}$ denotes the output node embedding of $l$-th layer, $\mathbf{e}_n^{(L_c)}$ denotes the final embedding $\mathbf{e}_n^c$ of the node, $\mathcal{N}_n^{(t)}$ denotes the node $n$'s neighbor nodes set in turn $t$ and $\mathbf{W}_c^{(l+1)} \in \mathbb{R}^{d\times d}$ are trainable parameters.

For the global graph $\mathcal{G}_g = (\mathcal{N}, \mathcal{E})$, where $\mathcal{N} = \mathcal{U}\cup\mathcal{V}\cup\mathcal{P}$ and $\mathcal{E} = \mathcal{E}_{u,v}\cup\mathcal{E}_{p,v}$, we employ a $L_g$-layer Global Graph Neural Network (GGNN)(Schlichtkrull et al., 2018; Chen et al., 2020, 2019) to extract long-term preferences of users, and global correlations of items and attribute instances as in Zhang et al. (2022): $\mathbf{s}_{u\sim v}^{(l+1)}(n) = \mathbf{b}_g^{(l+1)} + \sum_{i\in\mathcal{N}_{r_{u\sim v}}(n)} \frac{\mathbf{W}_g^{(l+1)}\mathbf{s}_i^{(l)}}{\sqrt{|\mathcal{N}_{r_{u\sim v}}(i)||\mathcal{N}_{r_{u\sim v}}(n)|}}$, where $\mathcal{N}_{r_{u\sim v}}(n)$ denotes the neighbor nodes of node $n$ with the edge type $r_{u\sim v}$, which denotes the user $u$ has interacted the item $v$. $\mathbf{W}_g^{(l+1)}$ and $\mathbf{b}_g^{(l+1)}$ are both trainable parameters. We can get

$\mathbf{s}_{p\sim v}^{(l+1)}(n)$ similarly, where the $r_{p\sim v}$ denotes that the item $v$ is associated with the attribute instance $p$. Then we can get node embeddings $\mathbf{s}_n^{(l+1)}$ of users, attributes and items respectively: $\mathbf{s}_u^{(l+1)} = \text{ReLU}(\mathbf{s}_{u\sim v}^{(l+1)}(n))$, $\mathbf{s}_p^{(l+1)} = \text{ReLU}(\mathbf{s}_{p\sim v}^{(l+1)}(n))$, $\mathbf{s}_v^{(l+1)} = \text{ReLU}(\text{mean}(\mathbf{s}_{u\sim v}^{(l+1)}(n), \mathbf{s}_{p\sim v}^{(l+1)}(n)))$. We denote the output of the last layer $\mathbf{s}_n^{(L_g)}$ as the final embedding $\mathbf{s}_n^g$ of the node.

**Hierarchical-Interest Extractor.** Following Zhang et al. (2022), we use a multi-attention mechanism to capture user's diverse interests. First, we fuse the embeddings of the rejected items and attribute instances to represent the negative interest of the user:

$$\mathbf{v}_{N+1} = \mathbf{W}_{rej}\Big(\frac{1}{|\mathcal{N}_{rej}|}\sum_{n\in\mathcal{N}_{rej}} \mathbf{s}_n^g\Big) \tag{4}$$

where $\mathcal{N}_{rej} = \mathcal{V}_{rej}\cup\mathcal{P}_{rej}$, and $\mathbf{W}_{rej} \in \mathbb{R}^{d\times d}$ are trainable parameters. Then based on the dynamic tree, we get the special labels $l$ for accepted attribute instance embeddings $[v_1, v_2, ..., v_N]$ and $v_{N+1}$:

$$\mathbf{l}_i = \begin{cases} [1, 0], & \text{if } p_i \in \mathcal{P}_{special\_acc} \\ [0, 1], & \text{otherwise} \end{cases} \tag{5}$$

where $i = 1, 2, ..., N, N + 1$. It is worth noting that we use the embedding $\mathbf{e}_n^c$ of the current graph $\mathcal{G}_u^{(t)}$ for $v_1$ to $v_N$ to balance the user's short-term and long-term preferences.

Finally, we utilize $K_I$ attention networks with $M$ iterations to adjust the weights between $[v_1, v_2, ..., v_N, v_{N+1}]$. The method for calculating the initial iteration of each attention network to obtain the interest embedding $\mathbf{q}_k^{(1)}$ is as follows:

$$\mathbf{q}_k^{(1)} = \sum_{n=1}^{N+1} \alpha_{k,n}^{(1)}\mathbf{v}_n, k \in \{1, \ldots K_I\} \tag{6}$$

$$\alpha_{k,n}^{(1)} = \frac{\exp\big(\mathbf{h}_k^\top\sigma\big(\mathbf{W}_k(\mathbf{v}_n\|\mathbf{l}_n)\big)\big)}{\sum_{n'=1}^{N+1}\exp\big(\mathbf{h}_k^\top\sigma\big(\mathbf{W}_k(\mathbf{v}_{n'}\|\mathbf{l}_{n'})\big)\big)} \tag{7}$$

where $\mathbf{h}_k$ and $\mathbf{W}_k$ are trainable parameters. Based on the $m-1$-th iteration results, we can get the $\mathbf{q}_k^{(m)}$ as follows:

$$\mathbf{q}_k^{(m)} = \sum_{n=1}^{N+1} \alpha_{k,n}^{(m)}\mathbf{v}_n \tag{8}$$

$$\alpha_{k,n}^{(m)} = \frac{\exp(\mathbf{h}_k^\top \sigma(\mathbf{W}_k(\mathbf{v}_n \| \mathbf{l}_n \| \mathbf{q}_k^{(m-1)}))}{\sum_{n'=1}^{N+1} \exp(\mathbf{h}_k^\top \sigma(\mathbf{W}_k(\mathbf{v}_{n'} \| + \alpha_{k,n}^{(m-1)}) \over \mathbf{l}_{n'} \| \mathbf{q}_k^{(m-1)})) + \alpha_{k,n'}^{(m-1)})} \quad (9)$$

We define the output of the $M$-th iteration as the final embeddings. These embeddings are represented by $\{\mathbf{q}_1^M, \mathbf{q}_2^M, ..., \mathbf{q}_{K_I}^M\}$, where $\mathbf{q}_1^M, \mathbf{q}_2^M, ..., \mathbf{q}_{K_I}^M$ are the embedded vectors that capture the hierarchical interests.

**Action Decision Policy Learning**. A good strategy not only makes the recommendation at the appropriate time but also exhibits flexible adaptation to users' feedback. Additionally, it maintains conversational topics and adapts to various scenarios to ensure user comfort throughout the interaction(Gao et al., 2021). Previous works focus on the strategy for determining *when to ask attributes or when to recommend items*, while our strategy also requires determining *which aspects to ask*. Therefore, we divide the decision-making process into two stages. In stage I, the system aims to identify the aspects that the user prefers, while in stage II, the system will ask about attributes related to these positive aspects or recommend items.

Following Deng et al. (2021); Zhang et al. (2022), we select top-$K_v$ items based on the recommendation score $w_v^{(t)}$ and top-$K_p$ attribute instances based on the entropy score as the action space $\mathcal{A}_t$. Inspired by Zhang et al. (2022), we adopt a dueling Q-network to determine the next action. The Q-value $Q(s_t, a_t)$ of the state $s_t$ and the action $a_t$ is calculated by: $Q(s_t, a_t) = \max_k (f_{\theta_V}(\mathbf{q}_k) + f_{\theta_A}(\mathbf{q}_k, a_t)), k \in \{1, \ldots K_I\}$, where $f_{\theta_V}(\cdot)$ and $f_{\theta_A}(\cdot)$ are two separate multi-layer perceptions (MLP). Following the Bellman equation (Bellman and Kalaba, 1957), the optimal Q-value $Q^*(s_t, a_t)$ is calculated by: $Q^*(s_t, a_t) = \mathbb{E}_{s_{t+1}}[r_t + \gamma \max_{a_{t+1} \in \mathcal{A}_{t+1}} Q^*(s_{t+1}, a_{t+1} | s_t, a_t)]$, where $r_t$ is the reward based on the user's feedback and $\gamma$ is the discounted factor.

In stage I, we adopt the sum-based strategy. For each candidate aspect node of the dynamic tree, we sum the Q-values of its child nodes in $\mathcal{A}_t$ to obtain the aspect-level score by which we select top-$K_{asp}$ aspect instances to ask. Once the user accepts some aspects, the dynamic tree no longer updates the first layer nodes and the system will transition from stage I to stage II. In stage II, we adopt the top-based strategy. The system first selects the action with the max Q-value. If the selected action points to an item, the system will recommend top-$K$ items. Otherwise, the system will ask top-$K_{att}$ attribute instances. Moreover, we define a replay buffer $D$ to store the experience $(s_t, a_t, r_t, s_{t+1}, \mathcal{A}_{t+1})$ and define a loss function:$\mathcal{L} = \mathbb{E}_{(s_a, a_t, r_t, s_{t+1}, \mathcal{A}_{t+1}) \sim \mathcal{D}}[(y_t - Q(s_t, a_t; \theta_Q, \theta_M))^2]$, where $\theta_M$ is the set of parameters to capture hierarchical-interest embeddings, $\theta_Q = \{\theta_V, \theta_A\}$, and $y_t$ is the target value, which is based on the optimal Q-function: $y_t = r_t + \gamma \max_{a_{t+1} \in \mathcal{A}_{t+1}} Q(s_{t+1}, a_{t+1}; \theta_Q, \theta_M)$.

We utilize the double DQN (Van Hasselt et al., 2016) method to combat the overestimation bias present in the original DQN. Specifically, we employ a periodic copy of a target network $Q'$ from the online network to train the model, in accordance with Deng et al. (2021); Zhou et al. (2020); Zhang et al. (2022).

## 5 Experiment

### 5.1 Datasets

Following Zhang et al. (2022), we conduct experiments on four benchmark datasets: Yelp, LastFM, Amazon-Book, and MovieLens. For each conversation episode, we sample $N_v$ items with partially overlapped attribute instances as acceptable items for the user.

### 5.2 Experiments Setup

**User Simulator**. We conduct a simulation of a conversation session for each observed user-item interaction pair $(u, \mathcal{V}_u)$. In this simulation, we consider each item $v_i \in \mathcal{V}_u$ as the target item that serves as the ground truth. To initiate the session, the simulated user specifies an attribute instance $p_0 \in \mathcal{P}_{same}$. Given a conversation, the simulated user's feedback of each turn follows the rules: (1) when the system asks a question, they will accept the aspects instances or attribute instances which are associated with any item in $\mathcal{V}_u$ and reject others; (2) when the system recommends a list of items, it will be accepted by the user if at least one item in $\mathcal{V}_u$ is included; (3) the user's patience will expire when the maximum number of turn $T$ is reached.

**Baselines**. To evaluate model performance, we compare our model with the following baselines: (1) Max Entropy(Lei et al., 2020a), which asks the user for an attribute based on the maximum entropy or recommends items with a certain probabil-

| Models | Yelp | | | LastFM | | | Amazon Book | | | MovieLens | | |
|---|---|---|---|---|---|---|---|---|---|---|---|---|
| | SR@15 | AT | hDCG | SR@15 | AT | hDCG | SR@15 | AT | hDCG | SR@15 | AT | hDCG |
| Abs Greedy | 0.222 | 13.48 | 0.073 | 0.635 | 8.66 | 0.267 | 0.189 | 13.43 | 0.089 | 0.273 | 12.19 | 0.138 |
| Max Entropy | 0.375 | 12.57 | 0.139 | 0.640 | 9.62 | 0.288 | 0.343 | 12.21 | 0.125 | 0.704 | 6.93 | 0.448 |
| CRM | 0.223 | 13.83 | 0.073 | 0.597 | 10.60 | 0.269 | 0.309 | 12.47 | 0.117 | 0.654 | 7.86 | 0.413 |
| EAR | 0.263 | 13.79 | 0.098 | 0.612 | 9.66 | 0.276 | 0.354 | 12.07 | 0.132 | 0.714 | 6.53 | 0.457 |
| SCPR | 0.413 | 12.45 | 0.149 | 0.751 | 8.52 | 0.339 | 0.428 | 11.50 | 0.159 | 0.812 | 4.03 | 0.547 |
| UNICORN | 0.438 | 12.28 | 0.151 | 0.843 | 7.25 | 0.363 | 0.466 | 11.24 | 0.170 | 0.836 | 3.82 | 0.576 |
| MCMIPL | 0.482 | 11.87 | 0.160 | 0.874 | **6.35** | **0.396** | 0.545 | 10.83 | 0.223 | 0.882 | **3.61** | **0.599** |
| **HutCRS** | **0.528**\* | **11.33**\* | **0.175**\* | **0.900**\* | 6.52 | 0.348 | **0.638**\* | **9.84**\* | **0.227**\* | **0.902**\* | 4.16 | 0.475 |

Table 1: Performance comparison of different models on the four datasets. * indicates statistically significant improvement (p < 0.01) over all baselines. hDCG stands for hDCG@(15, 10).

ity. (2) Abs Greedy(Christakopoulou et al., 2016), which only recommends items and updates itself. (3) CRM(Sun and Zhang, 2018), which is a single-round CRS that learns the policy deciding the next action. Following Lei et al. (2020a), we adapt CRM to MCR scenario. (4) EAR(Lei et al., 2020a), which proposes a three-stage model to better converse and recommend item to users. (5) SCPR(Lei et al., 2020b), which conducts interactive path reasoning on the graph to prune candidate attribute nodes in the graph, and employs the DQN to select an action. (6) UNICORN(Deng et al., 2021), which proposes a unified policy learning framework and adopts a dynamic weighted graph-based RL to select action. (7) MCMIPL(Zhang et al., 2022), which generates multiple choice questions, utilizes a union set strategy to select candidate items and the exact multi-interest of the user to select the next action. It is the state-of-the-art(SOTA) method.

**Parameters Setting** We randomly split each dataset into training, validation, and test sets in a 7:1.5:1.5 ratio. The embedding dimension is set to 64, and the batch size to 128. In stage I, We ask at most $K_{asp} = 4$ aspect instances in each turn. In stage II, we recommend top $K = 10$ items or ask $K_{att} = 2$ attribute instances in each turn. The maximum number of turn $T$ for a conversation is set to 15. We utilize the Adam optimizer with a learning rate of 1e-4. Discount factor $\gamma$ is set to be 0.999. Following Deng et al. (2021), we pretrain the node embeddings in the constructed KG with the training set using TransE (Bordes et al., 2013) via OpenKE (Han et al., 2018). We construct the global graph based on the training set. The numbers of current GNN layers $L_c$ and global GNN layers $L_g$ are set to 2 and 1, respectively. We ex-

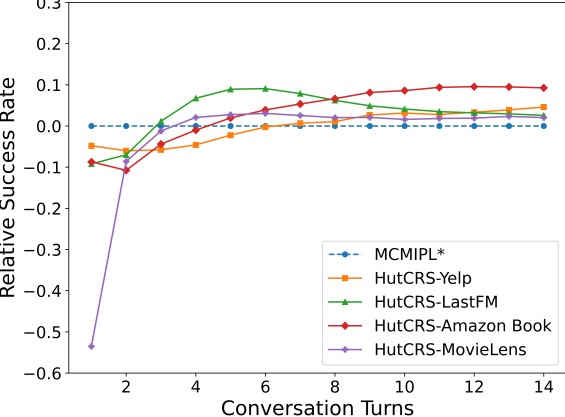

Figure 3: Comparisons at Different Conversation Turns.

tract the user's hierarchical interests with $K_I = 4$ attention networks, $M = 2$ iterations and the action space consists of $K_p = 10$ attribute instances and $K_v = 10$ items. The reward settings are as follows: $r_{rec\_suc} = 1$, $r_{rec\_fail} = -0.1$, $r_{ask\_att\_suc} = 0.01$, $r_{ask\_att\_fail} = -0.1$, $r_{ask\_asp\_suc} = 0.02$, $r_{ask\_asp\_fail} = -0.2$, $r_{quit} = -0.3$. Inspired by Zhang et al. (2022), we set the reward of asking a question as follows: In stage I, $r_t = \sum_{\mathcal{C}_{cur\_acc}^{(t)}} r_{ask\_asp\_suc} + \sum_{\mathcal{C}_{cur\_rej}^{(t)}} r_{ask\_asp\_fail}$, while in stage II, $r_t = \sum_{\mathcal{P}_{cur\_acc}^{(t)}} r_{ask\_att\_suc} + \sum_{\mathcal{P}_{cur\_rej}^{(t)}} r_{ask\_att\_fail}$. We set the maximum number $N_v$ of acceptable items as 2 and explore other settings in the hyper-parameter analysis.

**Evaluation Metrics**. Following Lei et al. (2020a); Deng et al. (2021); Zhang et al. (2022), we utilize the success rate at turn $t$ (SR@$t$) (Sun and Zhang, 2018) to measure the cumulative ratio of successful recommendation with the maximum turn $T$, and average turn (AT) to evaluate the average number

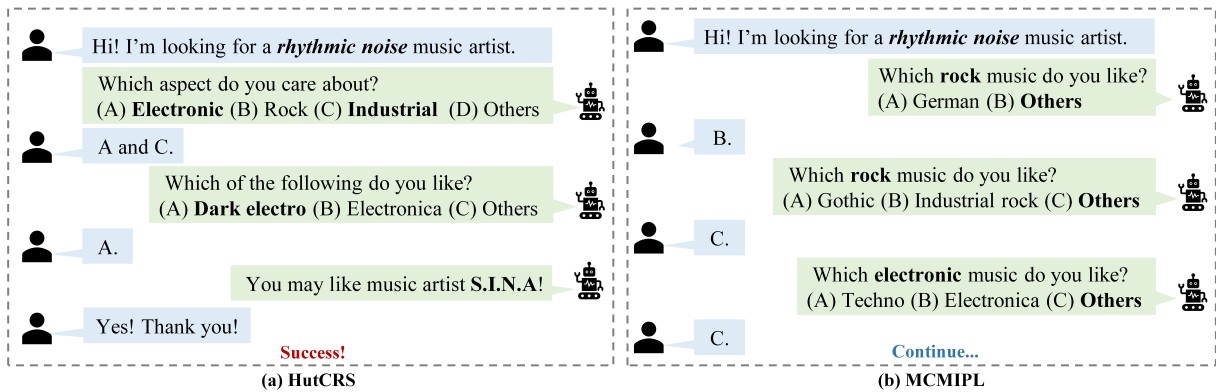

Figure 4: Sample conversations generated by HutCRS and MCMIPL.

of turns. Besides, we adopt hDCG@($T$,$K$) (Deng et al., 2021) to additionally evaluate the ranking performance of recommendations. Higher values of SR@$t$ and hDCG@($T$,$K$) indicate better performance, while a lower value of AT indicates overall higher efficiency.

## 5.3 Performance Comparison

The comparison experimental results of the baseline models and our models are shown in Table 1. In terms of SR@15, our model outperforms all the comparison methods with improvements of 10%, 3%, 17%, and 2% on four datasets. But in terms of AT and hDCG, our model only shows improvement on the Yelp and Amazon Book datasets. However, this result does not imply that our model is inefficient. We intuitively present the performance comparison of success rate at each turn in Figure 3. Relative success rate denotes the difference between HutCRS and the most competitive baseline MCMIPL, where the blue line of MCMIPL is set to $y = 0$ in the figure. We observe that the absence of recommendations in stage I for HutCRS leads to a lower success rate compared to MCMIPL in the initial turns. As a result, this negatively affects the model's performance on the AT and hDCG metrics, especially for relatively straightforward datasets such as lastfm and movielens. But for the more challenging Yelp and Amazon Book datasets, our model achieves improvement across all metrics. This shows that our method attains better performance than previous baselines.

## 5.4 Ablation Studies

In order to verify the effectiveness of some key designs, we conduct a series of ablation experiments on the Amazon-Book and MovieLens datasets. The results are shown in Table 2. We design four ab-

| Models | Amazon Book | | | MovieLens | | |
|---|---|---|---|---|---|---|
| | SR@15 | AT | hDCG | SR@15 | AT | hDCG |
| HutCRS | 0.638 | 9.84 | 0.227 | 0.902 | 4.16 | 0.475 |
| (a) | 0.630 | 9.85 | 0.225 | 0.890 | 4.25 | 0.467 |
| (b) | 0.619 | 10.00 | 0.221 | 0.898 | 4.26 | 0.468 |
| (c) | 0.615 | 10.05 | 0.218 | 0.871 | 4.77 | 0.432 |
| (d) | 0.541 | 10.79 | 0.189 | 0.882 | 4.69 | 0.438 |
| (e) | 0.629 | 9.86 | 0.227 | 0.875 | 4.42 | 0.444 |
| (f) | 0.592 | 10.06 | 0.217 | 0.878 | 4.34 | 0.468 |

Table 2: Results of the Ablation Study.

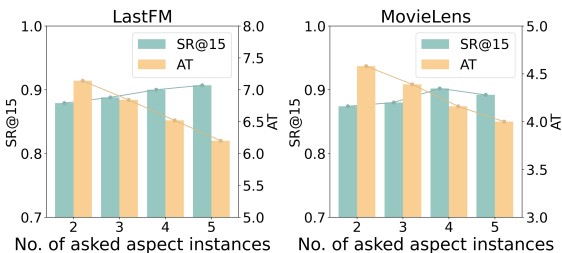

Figure 5: Performance comparisons w.r.t. $K_{asp}$

lation models: (a) removing the classification; (b) removing the global graph; (c) removing the current graph; (d) adopting binary questions in stage I; (e) replacing the sum-based strategy for selecting aspect instances with a random strategy; (f) replacing the top-based strategy for selecting attribute instances with a random strategy. All ablation models perform worse than HutCRS in terms of all metrics, which demonstrates the preeminence of HutCRS.

## 5.5 Hyper-parameter Analysis

Inquiring users with questions that incorporate a varying number of aspect instances ($K_{asp}$) influences the performance of the model. As illustrated

in Figure 5, the AT metric exhibits improvement when the value of $K_{asp}$ rises, whereas the SR@15 metric does not consistently display enhancement. This observation suggests that increasing $K_{asp}$ beyond a certain threshold may yield limited performance gains. The ongoing decline of the AT metric can be attributed to a reduction in the number of turns taken during stage I as $K_{asp}$ escalates.

## 5.6 Case Study

We further conduct qualitative analysis to demonstrate the role played by the hierarchical interest structure in successful recommendations. We randomly sample a real-world interaction from our model and the state-of-the-art method on LastFM based on the same test instance. The generated conversations by HutCRS and MCMIPL with the user simulator are presented in Figure 4. It can be observed that HutCRS first identifies the aspects that the user prefers, then asks about attributes related to the accepted aspects, and finally recommends the item. Compare to HutCRS, MCMIPL repeatedly asks about attributes of the aspect that the user is not interested in, resulting in lower efficiency. Therefore, the hierarchical interest structure makes conversations more user-friendly and explainable.

## Conclusion

In this paper, we propose a more realistic, user-friendly, and explainable framework HutCRS, which portrays the conversation as a hierarchical interest tree consisting of two stages. In addition, we design a HIPL module to integrate the decision-making process and classify attribute-level feedback to capture special information. Extensive experiments on four benchmark datasets demonstrate the superiority of our method.

## Limitations

While our model achieves a new state-of-the-art performance, it still has several limitations. Firstly, our framework employs a hierarchical interest tree structure, necessitating the inclusion of both attribute and aspect data in the dataset. Secondly, the conversation is divided into two stages. During the stage I, the system asks about aspects until the user accepts specific aspects and then proceeds to the stage II. Once in the stage II, the system is unable to revert to the stage I. Ideally, the agent should be able to recommend directly based on negative feedback regarding aspects in the stage I and return

to the stage I after entering the stage II, as this may enhance performance. Lastly, our method solely classifies feedback at the attribute level, lacking item-level classification, which also presents an opportunity for improvement.

## Ethics Statement

The data utilized in our study are sourced from open-access repositories, and do not pose any privacy concerns. We are confident that our research adheres to the ethical standards set forth by EMNLP.

## Acknowledgements

This work is supported in part by the National Key R&D Program of China under Grant No.2021ZD0111601; in part by the National Natural Science Foundation of China under Grant No.U21A20470, Grant No.61836012, and Grant No. 62206314; in part by the GuangDong Basic and Applied Basic Research Foundation under Grant No. 2023A1515011374 and Grant No. 2022A1515011835; in part by the Guangdong Province Key Laboratory of Information Security Technology; and in part by the China Postdoctoral Science Foundation Funded Project under Grant No. 2021M703687.

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
