# OpenReview forum: "HutCRS: Hierarchical User-Interest Tracking for Conversational Recommender System"
_EMNLP/2023/Conference — EMNLP 2023 Main_

### Official Review · Reviewer_9hoF · 2023-07-31

**Soundness:** 3

**Excitement:**

3: Ambivalent: It has merits (e.g., it reports state-of-the-art results, the idea is nice), but there are key weaknesses (e.g., it describes incremental work), and it can significantly benefit from another round of revision. However, I won't object to accepting it if my co-reviewers champion it.

**Missing References:**

1.	Is the process of constructing the full hierarchical interest tree and then pruning it to obtain a dynamic tree based on user feedback computationally expensive in terms of memory and time?

**Paper Topic And Main Contributions:**

Addressing the limitations of current attribute-based Conversational Recommender Systems (CRS) in accurately capturing user interest and reflecting real-world scenarios, the authors present a new framework named Hierarchical User-Interest Tracking for Conversational Recommender System (HutCRS) to enhance the user experience.  In HutCRS, the ongoing conversation is represented as a hierarchical interest tree, with the user initialized attribute, aspect instances, attribute instances, and items forming the root, first, second, and last layers respectively. The tree is dynamically pruned based on user feedback. Additionally, attribute-level feedback results are classified to capture special information effectively. To optimize the recommendation process, the authors employ a Hierarchical-Interest Policy Learning module to make decisions that lead to successful recommendations in fewer interactions. The proposed system's effectiveness is evaluated through extensive experiments conducted on four benchmark datasets.

**Reasons To Accept:**

1.	The authors recognize the limitations of current Multi-Choice Conversational Recommender (MCR) methods and enhance them to address a more realistic scenario, considering the user's authentic interest characteristics.
2.	In order to enhance the system's comprehension of users' preferences and requirements, the authors introduce the concept of a hierarchical interest tree, which accurately depicts users' interests and priorities.
3.	The authors extensively evaluate the proposed framework through rigorous experiments conducted on four benchmark datasets, providing compelling evidence of its effectiveness.


**Reasons To Reject:**

1.	The authors' construction of a hierarchical interest tree to represent user preferences in each session raises concerns about the time complexity of the overall process. To demonstrate the efficiency of the proposed framework, it would be beneficial if the paper includes a detailed analysis of the time complexity and compares it with other methods.
2.	Although the framework bears similarities to [1], the paper should emphasize and better showcase the improvements and advancements achieved by the proposed approach over [1]. This clarification would add more clarity and value to the research, enabling readers to understand the novel contributions of the proposed framework.
3.	For a comprehensive understanding of the proposed approach's performance, it would be valuable to include information on the success rate at each conversation turn. This data can offer insights into the framework's comparative performance throughout the recommendation process, helping readers to assess its effectiveness in a more nuanced manner.

[1] Yiming Zhang, Lingfei Wu, Qi Shen, Yitong Pang, Zhi-hua Wei, Fangli Xu, Bo Long, and Jian Pei. 2022. Multiple choice questions based multi-interest policy learning for conversational recommendation. In Proceedings of the ACM Web Conference 2022, pages 2153–2162.


**Reproducibility:**

3: Could reproduce the results with some difficulty. The settings of parameters are underspecified or subjectively determined; the training/evaluation data are not widely available.

**Reviewer Confidence:**

4: Quite sure. I tried to check the important points carefully. It's unlikely, though conceivable, that I missed something that should affect my ratings.

---

> ### Author Rebuttal · Authors · 2023-08-28
>
> Dear Reviewer,
>
> ​    ​    ​    ​    ​    ​    Thank you for reviewing our paper and providing valuable comments and suggestions. We sincerely appreciate your attention to our research and the time you have dedicated to it. Based on your feedback, we have engaged in thorough discussions and analysis, and we will make the necessary revisions to enhance the quality and accuracy of the paper. In the following response, we provide detailed explanations of our responses to each question and the specific measures taken for the revisions.
>
> **Comment1** The authors' construction of a hierarchical interest tree to represent user preferences in each session raises concerns about the time complexity of the overall process. To demonstrate the efficiency of the proposed framework, it would be beneficial if the paper includes a detailed analysis of the time complexity and compares it with other methods.
>
> **Response:** It is worth noting that our hierarchical interest tree hardly incurs additional time and memory overhead. Although we portray the conversation as a hierarchical interest tree, it is not necessary to store the full tree. It is enough to promote dialogue by saving the nodes (such as candidate attribute sets) and the relationships between nodes which can be obtained from the knowledge graph. In this regard, existing methods also store nodes (except for aspect nodes and special information) and make use of a knowledge graph, so our hierarchical interest tree is low-cost. Furthermore, compared to [1], our model only asks about attributes related to aspects that the user is interested in. This can reduce the size of the candidate attribute set and consequently decrease the number of scoring computations performed on attribute instances, making the conversation more efficient.
>
> Specifically, in a turn of scoring attribute instances, [1] has a time complexity of $O(n_1 \cdot t_1)$, while our model has a time complexity of $O(n_2 \cdot t_2)$, where $n$ represents the size of the candidate attribute set, and $t$ represents the time complexity of scoring a single attribute instance. It is worth noting that $t_1$ is approximately equal to $t_2$, and $n_1$ is approximately k times larger than $n_2$, where k represents the average factor by which $n_1$ is larger than $n_2$. Since the four datasets on average contain around 30 aspect instances, k is expected to be greater than 10. Therefore, our model is more efficient in comparison.
>
> **Comment2** Although the framework bears similarities to [1], the paper should emphasize and better showcase the improvements and advancements achieved by the proposed approach over [1]. This clarification would add more clarity and value to the research, enabling readers to understand the novel contributions of the proposed framework.
>
> **Response:** Thanks for your suggestion. We will emphasize and better showcase the improvements and advancements achieved by the proposed approach over [1]. Compared to [1], we have made several improvements in our dialogue process. Firstly, our system now starts by capturing the aspects that interest the user before asking for attributes or recommending items. This not only enhances user-friendliness but also ensures more authentic feedback at the attribute-level. Secondly, we have utilized a hierarchical interest tree to reduce the size of the candidate attribute set, resulting in a more efficient computation process. Additionally, we have classified the feedback at the attribute-level and incorporated special information into the embeddings, enabling the model to better understand user’s feedback with greater precision. Lastly, our model's overall performance outperforms [1], especially on datasets that are inherently more challenging for successful recommendations. This demonstrates the superior potential of our model.
>
> **Comment3** For a comprehensive understanding of the proposed approach's performance, it would be valuable to include information on the success rate at each conversation turn. This data can offer insights into the framework's comparative performance throughout the recommendation process, helping readers to assess its effectiveness in a more nuanced manner.
>
> **Response:** Thanks for your suggestion. In fact, our paper has already included Success Rate at turn $t$ (SR@15) metric where turn $t$ ranges from 1 to 15, which is a most popular metric to evaluate the models' ability to minimize number of conversation turns in conversational recommendation. Most researches [1] [2] [3] [4] [5] consider that SR@15 (i.e., $t \in [0, 15]$) can be effectively to evaluate the conversation task since a robust CRS could successfully recommend items in the shortest possible number of rounds, and thus $t \in [0, 15]$ contains enough turns. Besides, we will draw a figure to show the success rate at each conversation turn on four datasets to make it be more intuitively understood in the final paper.
>
> **Comment4** Is the process of constructing the full hierarchical interest tree and then pruning it to obtain a dynamic tree based on user feedback computationally expensive in terms of memory and time?
>
> **Response:** Our hierarchical interest tree is somewhat similar to the one mentioned in Paper [6]. We will include [6] in the list of references to help users better understand the hierarchical interest tree. However, as mentioned earlier, since we can directly retrieve the relationships between nodes from the knowledge graph, and both the knowledge graph and the information of the nodes themselves are also utilized in other methods, it hardly incurs any additional time and space overhead.
>
> [1] Yiming Zhang, Lingfei Wu, Qi Shen, Yitong Pang, Zhi-hua Wei, Fangli Xu, Bo Long, and Jian Pei. 2022. Multiple choice questions based multi-interest policy learning for conversational recommendation. In Proceedings of the ACM Web Conference 2022, pages 2153–2162.
>
> [2] Lei W, He X, Miao Y, et al. Estimation-action-reflection: Towards deep interaction between conversational and recommender systems[C]//Proceedings of the 13th International Conference on Web Search and Data Mining. 2020: 304-312.
>
> [3] Deng Y, Li Y, Sun F, et al. Unified conversational recommendation policy learning via graph-based reinforcement learning[C]//Proceedings of the 44th International ACM SIGIR Conference on Research and Development in Information Retrieval. 2021: 1431-1441.
>
> [4] Lei W, Zhang G, He X, et al. Interactive path reasoning on graph for conversational recommendation[C]//Proceedings of the 26th ACM SIGKDD international conference on knowledge discovery & data mining. 2020: 2073-2083.
>
> [5] Xu K, Yang J, Xu J, et al. Adapting user preference to online feedback in multi-round conversational recommendation[C]//Proceedings of the 14th ACM international conference on web search and data mining. 2021: 364-372.
>
> [6] Zhu H, Li X, Zhang P, et al. Learning tree-based deep model for recommender systems[C]//Proceedings of the 24th ACM SIGKDD International Conference on Knowledge Discovery & Data Mining. 2018: 1079-1088.

---

### Official Review · Reviewer_DXwT · 2023-08-01

**Soundness:** 3

**Excitement:**

3: Ambivalent: It has merits (e.g., it reports state-of-the-art results, the idea is nice), but there are key weaknesses (e.g., it describes incremental work), and it can significantly benefit from another round of revision. However, I won't object to accepting it if my co-reviewers champion it.

**Paper Topic And Main Contributions:**

This paper proposes HutCRS, a conversational recommender system using a hierarchical interest tree to model user preferences. The model consists of two stages: hierarchical user state tracking and hierarchical-interest policy learning.
Besides, the paper classifies the attribute-level feedback results to further enhance the system’s ability to comprehend the needs and preferences of users.
Experiments on four datasets show improvements over baselines.

**Reasons To Accept:**

* Classifying attribute feedback as special positive/negative is an interesting idea to capture non-obvious preferences that improve recommendation accuracy.
* The proposed method is more user-friendly, as it allows users to provide feedback on multiple aspects, unlike traditional systems that only allow single-item feedback.
* Experiments on four benchmarks show the effectiveness of the proposed method.

**Reasons To Reject:**

* The paper does not perform convincingly in empirical evaluations. Notably, it does not outperform the baseline MCMIPL on the LastFM and MovieLens datasets. This raises questions about the practicality and efficacy of the proposed method.
* It appears that the improvements seem to be more from allowing multiple choice feedback than the proposed hierarchical tree and policy learning contributions. More evidence is needed to demonstrate their effectiveness.
* The necessity of the current and global graphs in the ablation study is not well justified. There is a lack of evidence that these components significantly contribute to the system's performance.
* The connection between the proposed method and the challenges mentioned in the introduction is unclear. For example, the paper does not explain how the hierarchical tree framework handles situations where "users might not necessarily like all attributes of the target item, nor dislike attributes that are not included in the target item."


**Reproducibility:**

3: Could reproduce the results with some difficulty. The settings of parameters are underspecified or subjectively determined; the training/evaluation data are not widely available.

**Reviewer Confidence:**

3: Pretty sure, but there's a chance I missed something. Although I have a good feel for this area in general, I did not carefully check the paper's details, e.g., the math, experimental design, or novelty.

**Typos Grammar Style And Presentation Improvements:**

* The paper's clarity could be improved. For instance, Figure 2 lacks clarity in portraying the two-stage process, and Figure 3 is placed too far from the corresponding text.
* "Hrarchical" in line 093 should be "Hierarchical"

---

> ### Author Rebuttal · Authors · 2023-08-28
>
> Dear Reviewer,
>
> ​    ​    ​    ​    ​    ​    Thank you for reviewing our paper and providing valuable comments and suggestions. We sincerely appreciate your attention to our research and the time you have dedicated to it. Based on your feedback, we have engaged in thorough discussions and analysis, and we will make the necessary revisions to enhance the quality and accuracy of the paper. In the following response, we provide detailed explanations of our responses to each question and the specific measures taken for the revisions.
>
> **Comment1** The paper does not perform convincingly in empirical evaluations. Notably, it does not outperform the baseline MCMIPL on the LastFM and MovieLens datasets. This raises questions about the practicality and efficacy of the proposed method.
>
> **Response:** It is indeed true that our model does not outperform the baseline MCMIPL in terms of AT and hDCG on the LastFM and MovieLens datasets. However, as mentioned in lines 541-547 of the paper, this does not imply that our method is inefficient. Firstly, the primary objective of our work is to extend the existing CRS to a more realistic, user-friendly, and explainable setting. In this new and more challenging setting, our model still achieves better overall performance. Secondly, unlike MCMIPL, our model does not perform recommendations in stage I; instead, it invests additional turns to gather aspects of user interest. Intuitively, this necessitates more turns for our model to complete the recommendations, particularly for simpler datasets like LastFM and MovieLens, resulting in lower AT and hDCG metrics. Lastly, regarding SR@15, our model outperforms MCMIPL on four datasets, indicating that our model has better potential.
>
> **Comment2** It appears that the improvements seem to be more from allowing multiple choice feedback than the proposed hierarchical tree and policy learning contributions. More evidence is needed to demonstrate their effectiveness.
>
> **Response:** Thanks for your comment. We believe it is highly reasonable that adopting binary choice questions (i.e., yes/no) in stage I leads to the most significant decline in the model's performance. The purpose of stage I is to gather the aspects they are interested in, and only those positive users' feedback can be transformed into stage II for recommendations. Therefore, if the system can only inquire about one aspect instance each turn, it proves not only less user-friendly but also results in the system spending more turns in stage I, consequently leading to a noticeable decrease in the model's performance. We conduct statistical analysis on the Amazon Book dataset. If the system only asks about one aspect in each turn, the average number of turns required in stage I would increase by approximately 37\%.
>
> **Comment3** The necessity of the current and global graphs in the ablation study is not well justified. There is a lack of evidence that these components significantly contribute to the system's performance.
>
> **Response:** Thank you for your suggestions. We conduct ablation studies on the Amazon Book and MovieLens datasets to investigate the effects of the global graph and the current graph. The experimental results are presented in the table below:
> | model         | datasets      | SR@15 | AT   | hDCG  |
> |---------------|---------------|-------|------|-------|
> | HutCRS        | Amazon Book   | 0.638 | 9.84 | 0.227 |
> | -global graph | Amazon Book   | 0.619 | 10.00| 0.221 |
> | -current graph| Amazon Book   | 0.615 | 10.05| 0.218 |
> | HutCRS        | MovieLens     | 0.902 | 4.16 | 0.475 |
> | -global graph | MovieLens     | 0.898 | 4.26 | 0.468 |
> | -current graph| MovieLens     | 0.871 | 4.77 | 0.432 |
>
> As can be seen from the table, both the global graph and the current graph have a positive impact on the model's performance, with the current graph having a relatively larger influence.
>
> **Comment4** The connection between the proposed method and the challenges mentioned in the introduction is unclear. For example, the paper does not explain how the hierarchical tree framework handles situations where "users might not necessarily like all attributes of the target item, nor dislike attributes that are not included in the target item."
>
> **Response:** Thanks for your suggestion. As mentioned in lines 105-111 of the paper, we first allow users to select the aspects they are interested in, and then ask about relevant attribute instances. We believe that users have clear preferences for these attribute instances: if an attribute instance belongs to the target item and falls within the aspects the user is interested in, the user will accept it. We will emphasize it in our paper later.
>
> **Comment5** The paper's clarity could be improved. For instance, Figure 2 lacks clarity in portraying the two-stage process, and Figure 3 is placed too far from the corresponding text.
>
> **Response:** Thanks for your suggestion. We will polish our paper and the layout of the figures accordingly.
>
> **Comment6** "Hrarchical" in line 093 should be "Hierarchical".
>
> **Response:** Thanks for pointing out the errors. We will correct these errors in the final paper.

---

### Official Review · Reviewer_2e3F · 2023-08-03

**Soundness:** 3

**Excitement:**

3: Ambivalent: It has merits (e.g., it reports state-of-the-art results, the idea is nice), but there are key weaknesses (e.g., it describes incremental work), and it can significantly benefit from another round of revision. However, I won't object to accepting it if my co-reviewers champion it.

**Paper Topic And Main Contributions:**

The paper's main topic is a Conversational Recommender System (CRS) that utilizes a hierarchical interest tree and attribute-level feedback to enhance the system's ability to comprehend the needs and preferences of users. The system is designed to generate a personalized recommendation for each user through a multi-turn dialogue process.

The main contributions of the paper are:
1. The proposal of a hierarchical interest tree and the utilization of the classification results of attribute-level feedback to further enhance the system's ability to comprehend the needs and preferences of users.
2. Demonstrating the superiority of HutCRS compared with state-of-the-art baselines through extensive experiments on four benchmarks, including Yelp, LastFM, Amazon-Book, and MovieLen datasets.

**Reasons To Accept:**

1.  HutCRS introduces interest-based policy learning and a hierarchical interest tree approach to dialog modeling, allowing the system to understand users' interests and preferences better and dynamically adjust based on user feedback.
2.  The experimental results show that HutCRS has higher recommendation accuracy as well as shorter conversation lengths compared to the traditional conversational recommender system approach.
3.  HutCRS introduces a deep reinforcement learning-based conversation management approach that allows the system to make more intelligent conversation management and recommendation decisions.

**Reasons To Reject:**

1. The authors should conduct a t-test to verify the proposed model's effectiveness, as the reinforcement learning technique is hard to implement.
2.  The task of conversational recommendation is still interesting, but I feel that this task is still difficult to evaluate because of the lack of real datasets. It would be nice if the authors could try to make an effort to build real datasets.
3.  The method proposed in the article is still complex and open source code is recommended.

**Reproducibility:**

2: Would be hard pressed to reproduce the results. The contribution depends on data that are simply not available outside the author's institution or consortium; not enough details are provided.

**Reviewer Confidence:**

3: Pretty sure, but there's a chance I missed something. Although I have a good feel for this area in general, I did not carefully check the paper's details, e.g., the math, experimental design, or novelty.

---

> ### Author Rebuttal · Authors · 2023-08-28
>
> Dear Reviewer,
>
> ​    ​    ​    ​    ​    ​    Thank you for reviewing our paper and providing valuable comments and suggestions. We sincerely appreciate your attention to our research and the time you have dedicated to it. Based on your feedback, we have engaged in thorough discussions and analysis, and we will make the necessary revisions to enhance the quality and accuracy of the paper. In the following response, we provide detailed explanations of our responses to each question and the specific measures taken for the revisions.
>
> **Comment1** The authors should conduct a t-test to verify the proposed model's effectiveness, as the reinforcement learning technique is hard to implement.
>
> **Response:** Thanks for your suggestion. We have conducted one-sample t-tests on all four datasets with a sample size of 40. The results indicate a significant difference between our model's performance and the strongest baseline MCMIPL (p < 0.01). Besides, we will provide the t-test on all datasets in the final paper.
>
> **Comment2** The task of conversational recommendation is still interesting, but I feel that this task is still difficult to evaluate because of the lack of real datasets. It would be nice if the authors could try to make an effort to build real datasets.
>
> **Response:** Thanks for your suggestion. We truly appreciate the value of using real datasets for evaluating this task, as the existing works mostly rely on the user simulator to simulate real users, which introduces biases. Therefore, in this study, we aim to extend the existing CRS to a more realistic, user-friendly, and explainable setting to ensure the effectiveness of the evaluation as much as possible. In the future, we will make an effort to construct real datasets.
>
> **Comment3** The method proposed in the article is still complex and open-source code is recommended.
>
> **Response:**  Thanks for your suggestion. We will release our code and model.

---

### Meta-Review · Area_Chair_karq · 2023-09-13

**Recommendation:** 3

**Metareview:**

The paper proposes an approach to produce personalized recommendation through multi-turn dialogue. User interests are represented hierarchically through a tree during dialogue, which is pruned and used to make recommendations (in a policy learning phase).

**Pros**: Reviewers agree the paper introduces new concepts (hierarchical interest tree and associated policy learning) that provide practical benefit to the to the conversational recommendation problem. Some reviewers note the solution has "interesting" new ideas, is more "user friendly", and has extensive evaluation (4 benchmarks).

**Cons**: Most reviewers raised technical concerns including: missing ablations, the use of simulated data, statistical significance of results, consistency of improvements compared to baselines as well as some concerns about motivation/hypotheses. Still, during the rebuttal authors appear to address many of these concerns. All reviewers acknowledge these responses post-rebuttal, and no review indicates serious technical concerns in the scoring. At the same time, most reviewers are ambivalent - there is no champion among the reviewers.

Overall a fairly middle ground paper: reviewers technical concerns are not serious enough to warrant low scores, and excitement among reviewers is not high enough to elicit a champion.

---

### Decision · Program_Chairs · 2023-10-07

**Decision:**

Accept-Main

**Comment:**

The paper proposes an approach to produce personalized recommendation through multi-turn dialogue. User interests are represented hierarchically through a tree during dialogue, which is pruned and used to make recommendations (in a policy learning phase).

**Pros**: Reviewers agree the paper introduces new concepts (hierarchical interest tree and associated policy learning) that provide practical benefit to the to the conversational recommendation problem. Some reviewers note the solution has "interesting" new ideas, is more "user friendly", and has extensive evaluation (4 benchmarks).

**Cons**: Most reviewers raised technical concerns including: missing ablations, the use of simulated data, statistical significance of results, consistency of improvements compared to baselines as well as some concerns about motivation/hypotheses. Still, during the rebuttal authors appear to address many of these concerns. All reviewers acknowledge these responses post-rebuttal, and no review indicates serious technical concerns in the scoring. At the same time, most reviewers are ambivalent - there is no champion among the reviewers.

Overall a fairly middle ground paper: reviewers technical concerns are not serious enough to warrant low scores, and excitement among reviewers is not high enough to elicit a champion.